METHODS

# Compression-based inference of network motif sets

**Alexis Bénichou**[1,2]*, **Jean-Baptiste Masson**[1,2], **Christian L. Vestergaard**[1,2]*

**1** Institut Pasteur, Université Paris Cité, CNRS UMR 3751, Decision and Bayesian Computation, Paris, France, **2** Epiméthée, Inria, Paris, France

* alexis.benichou@pasteur.fr (AB); christian.vestergaard@cnrs.fr (CLV)

**Data Availability Statement:** All data used are publicly available and are referenced in Table 2. Data files can be found at https://gitlab.pasteur.fr/sincobe/brain-motifs/-/tree/master/data. All scripts used to generate figures can be found at https://gitlab.pasteur.fr/sincobe/brain-motifs.

## Abstract

Physical and functional constraints on biological networks lead to complex topological patterns across multiple scales in their organization. A particular type of higher-order network feature that has received considerable interest is network motifs, defined as statistically regular subgraphs. These may implement fundamental logical and computational circuits and are referred to as "building blocks of complex networks". Their well-defined structures and small sizes also enable the testing of their functions in synthetic and natural biological experiments. Here, we develop a framework for motif mining based on lossless network compression using subgraph contractions. This provides an alternative definition of motif significance which allows us to compare different motifs and select the collectively most significant set of motifs as well as other prominent network features in terms of their combined compression of the network. Our approach inherently accounts for multiple testing and correlations between subgraphs and does not rely on *a priori* specification of an appropriate null model. It thus overcomes common problems in hypothesis testing-based motif analysis and guarantees robust statistical inference. We validate our methodology on numerical data and then apply it on synaptic-resolution biological neural networks, as a medium for comparative connectomics, by evaluating their respective compressibility and characterize their inferred circuit motifs.

## Author summary

Networks provide a useful abstraction to study complex systems by focusing on the interplay of the units composing a system rather than on their individual function. Network theory has proven particularly powerful for unraveling how the structure of connections in biological networks influence the way they may process and relay information in a variety of systems ranging from the microscopic scale of biochemical processes in cells to the macroscopic scales of social and ecological networks. Of particular interest are small stereotyped circuits in such networks, termed *motifs*, which may correspond to building blocks implementing fundamental operations, e.g., logic gates or filters. We here present a new tool that finds sets of motifs in networks based on an information-theoretic measure of how much they allow to compress the network. This approach allows us to evaluate the

**Funding:** This study was funded by L'Agence Nationale de la Recherche (SiNCoBe, ANR-20-CE45-0021 to CLV) and the "Investissements d'avenir" program under management of Agence Nationale de la Recherche, reference ANR-19-P3IA-0001 (PRAIRIE 3IA Institute) to AB, JBM, and CLV. The funders had no role in study design, data collection and analysis, decision to publish, or preparation of the manuscript.

**Competing interests:** The authors have declared that no competing interests exist.

collective significance of sets of motifs, as opposed to only individual motifs. We apply our methodology to compare the neural wiring diagrams, termed "connectomes", of the tadpole larva *Ciona intestinalis*, the ragworm *Platynereis dumerelii*, and the nematode *Caenorhabditis elegans* and the fruitfly *Drosophila melanogaster* at different developmental stages.

## Introduction

Network theory has highlighted remarkable topological features of many biological and social networks [1–3]. Some of the main ones are the *small world* property [4–7], which refers to a simultaneous high local clustering of connections and short global distances between nodes; scale-free features, most notably witnessed by a broad distribution of node degrees [8–11]; mesoscopic, and in particular modular, structuring [12–14]; and higher-order topological features [15], such as a statistical over-representation of certain types of subgraphs, termed *network motifs* [16–18].

We here focus on network motifs. They were first introduced to study local structures in social networks [19–21]. In biological networks, they are hypothesized to capture functional subunits (e.g., logic gates or filters) and have been extensively studied in systems ranging from transcription and protein networks to brain and ecological networks [2, 16–18, 22–24]. In contrast to most other remarkable features of biological networks, the well-defined structure and small size of network motifs mean that their function may be probed experimentally, both in natural [25, 26] and in synthetic experiments [25].

The prevailing approach to network motif inference involves counting or estimating the frequency of each subgraph type, termed a *graphlet*, and comparing it to its frequency in random networks generated by a predefined null model. Subgraphs that appear significantly more frequently in the empirical network than in the random networks are deemed motifs. While this procedure has offered valuable insights, it also suffers from several fundamental limitations which can make it statistically unreliable [27–32] (see S1 Text for an overview). Additionally, a flaw of testing-based approaches is that they cannot compare the significance of different motifs. Candidate motifs are usually treated independently. With increasingly richer and larger datasets, such methods thus risk detecting an exceedingly large amount of motifs (see, e.g., S1 Fig), which defies the original intent behind motif analysis as a means to capture essential, low-dimensional, mesoscopic properties of a network.

Information theory tells us that the presence of statistical regularities in a network makes it compressible [33]. Inspired by this fact, we here propose a methodology, based on lossless compression [34] as a measure of significance, that implicitly defines a generative model through the correspondence between universal codes and probability distributions [33, 35]. Through the minimum description length (MDL) principle [35, 36], our method infers the set of most significant motifs, as well as other node- and edge-level features, such as node degrees and edge reciprocity, by measuring how much they collectively allow to compress the network. We demonstrate how this approach allows to address the shortcomings of hypothesis testing-based motif inference. First, it naturally lets us account for multiple testing and correlations between different motifs. Furthermore, we can evaluate and compare even highly significant collections of motifs. Finally, our method selects not only the most significant motif configuration, but also node- and edge-level features, without needing to select the null model beforehand.

We first validate our approach on numerically generated networks with known absence or presence of motifs. We then apply our methodology to discover microcircuit motifs in synapse-resolution neuron wiring diagrams, the *connectomes*, of small animals which have recently become available thanks to advances in electron microscopy techniques and image segmentation [37–40]. We compare the compressibility induced by motif sets and other network features found in different brain regions of different animals. We namely analyze the connectome of *Caenorhabditis elegans* at different developmental stages, and the connectomes of different brain regions of both larval and adult *Drosophila melanogaster*, in addition to the complete connectomes of *Platynereis dumerelii* and larval *Ciona intestinalis*. We stress the exhaustive aspect of this diverse dataset: these constitute *all* the animals for which the complete anatomical, microscale wiring diagrams have presently been mapped. We find that all the connectomes are compressible, implying significant non-random structure. We find that the compressibility varies between connectomes, with larger connectomes generally being more compressible. We infer motif sets in most connectomes, but we do not find significant evidence for motifs in several of the smaller connectomes. The typical motifs tend to be dense subgraphs. We compare several topological measures of the motif sets, which show high similarity between connectomes, although with some significant differences.

## Materials and methods

In this section, we develop our methodology for compression-based inference of network motif sets. In "Graphlets and motifs", we first brush up on graph theory basics. In "Subgraph census", we describe the subgraph census procedure deployed to list subgraph occurrences. In "Compression, model selection, and hypothesis testing", we briefly review the MDL principle for model selection based on lossless compression. In "Graph compression based on subgraph contractions" we develop our code, corresponding to a probabilistic model, for network motif inference using subgraph contractions. In particular, we model a network with a prescribed motif set as an *expanded latent graph*, where expansion points designate the subset of latent nodes that embody motifs. In "Base codes and null models" we list the codes supporting the latent graph description, as well as codes providing purely dyadic representations. The latter serve as references that allow to quantify the significance of motif sets as compared to their respective best-fitting motif-free null model. In "Optimization algorithm" we describe our stochastic greedy optimization algorithm for selecting motif sets. Finally, in "Datasets" we present the artificial networks used for numerical validation and the neural connectomes that serve as real-world applications of our motif-based inference framework. All code and scripts are publicly available at gitlab.pasteur.fr/sincobe/brain-motifs.

## Graphlets and motifs

Network motif analysis is concerned with the discovery of statistically significant classes of subgraphs in empirically recorded graphs. We here restrict ourselves to directed unweighted graphs, but the concepts apply similarly to undirected graphs and may be extended to weighted graphs [41, 42], time-evolving and multilayer graphs [43–46], and hypergraphs [47, 48]. As is usual in motif analysis, we consider weakly connected subgraphs [16, 25]. This ensures that the subgraph may represent a functional subunit where all nodes can participate in information processing.

Let $G = (\mathcal{N}, \mathcal{E})$ denote the directed graph we want to analyze. For simplicity in comparing different representations of $G$, we consider $G$ to be node-labeled. Thus, the nodes $\mathcal{N} = (1, 2, \ldots, N)$ constitute an ordered set. The set of edges, $\mathcal{E} \subseteq \mathcal{N} \times \mathcal{N}$ indicates how the nodes are connected. By convention, a link $(i, j) \in \mathcal{E}$ indicates that $i$ connects to $j$. Note that, since $G$

is directed, the presence of $(i, j) \in \mathcal{E}$ does not imply the existence of $(j, i) \in \mathcal{E}$. We denote by $E = |\mathcal{E}|$ the number of edges. We only consider network data that form *simple* directed graphs, where $\mathcal{E}$ does not contain repeated elements: this is the definition of a set. The model we propose, however, makes use of *multigraphs* where $\mathcal{E}$ is a multi-set, which may contain repetitions.

A standard representation of a graph's connectivity is its *adjacency matrix*—denoted **A**—with entries given by $A_{ij} = |\{(i', j') \in \mathcal{E} : (i', j') = (i, j)\}|$. If the graph is simple, then the adjacency matrix is boolean, i.e., $A_{ij} = 1$ if $(i, j) \in \mathcal{E}$, otherwise $A_{ij} = 0$. When dealing with a multigraph, entries of the adjacency matrix take non-negative integer values, i.e., $A_{ij} \geq 1$ if $(i, j) \in \mathcal{E}$.

An *induced subgraph* $g = (v, \epsilon)$ of $G$ is the graph formed by a given subset $v \in \mathcal{N}$ of the nodes of $G$ and all the edges $\epsilon = \{(i, j) : i, j \in v \wedge (i, j) \in \mathcal{E}\}$ connecting these nodes in $G$.

An undirected graph $G_{\mathrm{un}}$ is called *connected* if there exists a path between all pairs of nodes in $G_{\mathrm{un}}$. A directed graph $G$ is *weakly connected* if the undirected graph obtained by replacing all the directed edges in $G$ with undirected ones is connected.

Two graphs $g = (v, \epsilon)$ and $g' = (v', \epsilon')$ are isomorphic if there exists a permutation $\sigma$ of the node indices of $g'$, such that the edges in the graphs perfectly overlap, i.e., $(i, j) \in \epsilon$ if and only if $(\sigma(i), \sigma(j)) \in \epsilon'$. A *graphlet*, denoted by $\alpha$, is an isomorphism class of weakly connected, induced subgraphs [49], i.e., the set $\alpha = \{g : g \cong g_\alpha\}$ of all graphs that are isomorphic to a given graph, $g_\alpha$.

Finally, a *motif* is a graphlet that is statistically significant. Traditionally, a significant graphlet is defined as one whose number of occurrences in $G$ is significantly higher than in random graphs generated by a null model [16]. Instead, we propose a method that selects a set of graphlets based on how well they allow to compress $G$. This lets us treat motif mining as a model selection problem through the MDL principle as we detail below.

## Subgraph census

The first step of a motif inference procedure is to perform a *subgraph census*, consisting in counting the graphlet occurrences. Subgraph census is computationally hard and many methods have been developed to tackle it [50].

For graphs with a small number of nodes, e.g., hundreds of nodes, we implemented the parallelized FaSe algorithm [51], while for larger graphs, i.e., comprising a thousand nodes or more, we rely on its stochastic version, Rand-FaSe [52]. The algorithms use Wernicke's ESU method (or Rand-ESU for large graphs) [53] for counting graphlet occurrences. It employs a trie data structure, termed *g-trie* [54], to store the graphlet labels in order to minimize the number of computationally costly subgraph isomorphism checks.

Since our algorithm relies on contracting individual subgraphs, we also need to store the location of each subgraph in $G$. Due to the large number of subgraphs, the space required to store this information may exceed working memory for larger graphs or graphlets (see discussion in S2 Text). Our most computationally challenging application—inference of motifs amongst all 3- to 5-node graphlets in the right mushroom body of the adult *Drosophila* connectome—requires storing 1.3 TB of data. In such cases, we write heavy textfiles of subgraph lists, one per graphlet, on the computer static memory, which are then retrieved individually from disk, at inference time (see S3 Text).

All scripts were run on the HPC cluster of the Institut Pasteur, but the less computationally challenging problem of inferring 3- to 4-node motifs can be run on a local workstation (see S2 Text).

## Compression, model selection, and hypothesis testing

The massive number of possible graphlet combinations and the correlations between graphlet counts within a network make classic hypothesis testing-based approaches for motif mining ill-suited for discovering motif sets. For example, there are approximately 10 000 different five-node graphlets and exponentially more possible combinations of such graphlets, making multiplicity a critical problem for hypothesis testing. Additionally, these approaches define motif significance by comparison with a random graph null model, and the results may depend on the choice of null model [27, 29] (see "Numerical validation" in the results below). In the context of motif mining, this choice can lead to ambiguities [27, 29, 30], thus rendering the analysis unreliable.

To address these problems, we cast motif mining as a model selection problem. We wish to select as motifs the multiset of graphlets, $\mathcal{S}^* = [\alpha^*]$ that, together with a tractable dyadic graph model, provides the most adequate model for $G$. The minimum description length (MDL) principle [35] states that, within an inductive inference framework with finite data, the most faithful representation of the observed system is given by the model that leads to the highest compression of the data—that is, of *minimum codelength*. It relies on an equivalence between codelengths and probabilities [33] and formalizes the well-known Occam's razor, or principle of parsimony. It is similar to Bayesian model selection and can be seen as a generalization of it [36].

To each dataset, model and parameter values, we associate a unique code, i.e., a label that identifies one representation. The code should be lossless, which means full reconstruction of the data from the compressed representation is possible [33, 35]. In practice, we are not interested in finding an actual code, but only in calculating the codelength of an optimal code [33], corresponding to our model.

Suppose we know the generative probability distribution of $G$, $P_\theta$, parameterized by $\theta$. Then, we can encode $G$ using a code whose length is equal to the negative log-likelihood [35],

$$L_\theta(G) = -\log P_\theta(G), \tag{1}$$

where log denotes the base-2 logarithm. (Note that an actual code would be between 1 to 2 bits longer than Eq (1) since real codewords are integer-valued and not continuous [35]). When the correct model and its parameters are unknown beforehand, we must encode both the model and the graph. To do this, we consider two-part codes, and, more generally, multi-part codes (see below). In a two-part code, we first encode the model and its parameters, using $L(\theta)$ bits, and then encode the data, $G$, conditioned on this model, using $-\log P_\theta(G)$ bits. This results in a total codelength of

$$L(G, \theta) = -\log P_\theta(G) + L(\theta). \tag{2}$$

With multi-part codes, we encode a hierarchical model following the same schema, where we first encode the model, then encode latent variables conditioned on the model, and then encode the data conditioned on the latent variables and the model.

When performing model selection, we consider a predefined set of models, $\mathcal{M} = \{P_\theta : \theta \in \Theta\}$, and we look for the one that, in an information-theoretic sense, best describes $G$. Following the MDL principle we select the parametrization $\theta^* \in \Theta$ that minimizes the description length,

$$\theta^* = \text{argmin}_{\theta \in \Theta} L(G, \theta). \tag{3}$$

Note that the second term in Eq (2), $L(\theta)$, quantifies the *model complexity*, which measures, in bits, the volume for storing the model parameters—this is a lossless encoding. Thus, one

must strike a balance between model likelihood and model complexity to minimize the description length, inherently penalizing overfitting.

While we focus on model selection, we also provide the absolute compression of the optimal model as an indicator of statistical significance. The link between compression and statistical significance is based on the *no-hypercompression inequality* [35]. It states that the probability that a given model, different from the true generating model, compresses the data more than the true model is exponentially small in the codelength difference. Formally, given a dataset $G$ (e.g., a graph) drawn from the distribution $P_0$ and another description $P_\theta$, then

$$P_0[-\log P_0(G) + \log P_\theta(G) \geq K] \leq 2^{-K}. \tag{4}$$

By identifying $P_0$ with a null model and $P_\theta$ with an alternative model, the no-hypercompression inequality thus provides an upper bound on the $p$-value, i.e., $p \leq 2^{-K}$. Note, however, that the above relation is not guaranteed to be conservative for composite null models (such as the configuration models that we consider below) [36, 55].

### Graph compression based on subgraph contractions

In practice, we compress the input graph by iteratively performing subgraph contractions each chosen from a set of possible graphlets, extending the approach of Bloem and de Rooij [34] which focused on a single graphlet. The model describes $G$ by a reduced representation, a multigraph $H$, with $N(H) < N(G)$ and $E(H) < E(G)$, in which a subset $\mathcal{V} \subseteq \mathcal{N}(H)$ of nodes are marked as *supernodes*, each formed by contracting a subgraph of $G$ into a single node (Fig 1A).

We let $\Gamma$ designate a predefined set of graphlets, which is the set of all graphlets we are interested in. In the following, we will generally consider all graphlets from three to five nodes—in which case $|\Gamma| = 9579$—but any predefined set of graphlets, or even a single graphlet, may be used. We define $\mathcal{S} = [\alpha]$ as a multiset of graphlets, corresponding to the subgraphs in $G$ that we contracted to obtain $H$. We define $\mathcal{A} = \{\alpha\}$ as the set containing the unique elements of $\mathcal{S}$ and let $m_\alpha = |[\beta \in \mathcal{S} : \beta = \alpha]|$ be the number of repetitions of $\alpha$ in $\mathcal{S}$. We finally let $P_\phi$ designate a dyadic random graph model, which is used to encode $H$. We consider four possible such *base* models (see Fig 1B and "Base codes and null models" below).

The full set of parameters and latent variables of our model is $\theta = \{H, \phi, \mathcal{S}, \mathcal{V}, \Gamma\}$, and its codelength can be decomposed into four terms,

$$L(G, \theta) = L(\Gamma, \mathcal{S}) + L(H, \phi) + L(\mathcal{V}|H, \mathcal{S}) + L(G|H, \mathcal{V}, \mathcal{S}, \Gamma) \tag{5}$$

where (i) $L(\Gamma, \mathcal{S})$ is the codelength for encoding the motif set; (ii) $L(H, \phi)$ is the codelength needed to encode the reduced multigraph $H$ using a base code corresponding to $P_\phi$; (iii) $L(\mathcal{V}|H, \mathcal{S})$ accounts for encoding which nodes of $H$ are supernodes and to which graphlets they correspond (i.e., their colors, Fig 1A); (iv) $L(G|H, \mathcal{V}, \mathcal{S}, \Gamma)$ corresponds to the information needed to reconstruct $G$ from $H$ (node labels, the orientations of the contracted subgraphs, and how the subgraph's nodes are wired to their respective external neighborhoods, see Fig 1C and 1D). We detail each of the four terms in turn.

The first term in Eq (5), $L(\Gamma, \mathcal{S})$ is given by

$$L(\Gamma, \mathcal{S}) = \sum_{\alpha \in \mathcal{A}} \log |\Gamma| + L_\mathbb{N}(|\Gamma|) + \sum_{\alpha \in \mathcal{A}} \log m_\mathrm{max} + L_\mathbb{N}(m_\mathrm{max}), \tag{6}$$

where $m_\mathrm{max} = \max_{\alpha \in \mathcal{A}} m_\alpha$ is the maximal number of repetitions of any of the graphlets in $\mathcal{A}$, and $L_\mathbb{N}(n) = \log[n(n+1)]$ is the codelength needed to encode an integer [35]. The first term in Eq (6) is the codelength needed to encode the identity of each inferred motif. Since there are $|\Gamma|$ possible graphlets, this requires $\log |\Gamma|$ bits per motif. The second term is the cost of

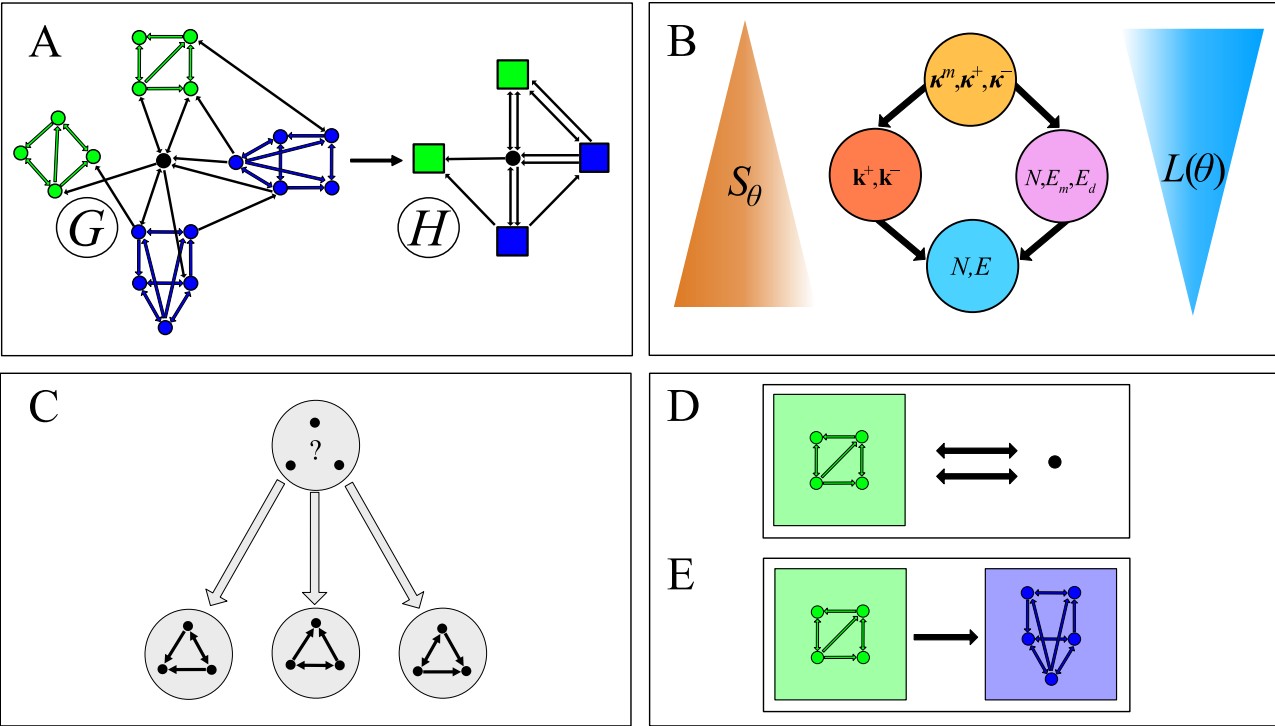

**Fig 1. Graphlet-based graph compression.** (A) Reduced representation of a graph $G$ obtained by contracting subgraphs into colored *supernodes* representing the subgraphs. (In this example, two different graphlets, colored in blue and green, are selected) The cost for encoding the reduced representation can be split into two parts: (i) encoding the multigraph $H$ obtained by contracting subgraphs in $G$, $L(H, \phi)$ (See "Base codes and null models" section), and (ii) encoding which nodes in $H$ are supernodes and their color, designating which graphlet they represent, $L(\mathcal{V}|H, \mathcal{S})$ [Eq (7)]. (B) Hierarchy of the four different dyadic graph models [56] used as base codes. Each node in the diagram represents a model. An edge between two nodes indicates that the upper model is less random than the lower. The models are: the Erdős-Rényi model $P_{(N,E)}$ (cyan); the directed configuration model $P_{(\mathbf{k}^+,\mathbf{k}^-)}$ (orange); the reciprocal Erdős-Rényi model $P_{(N,E_m,E_d)}$ (pink); and the reciprocal configuration model $P_{(\boldsymbol{\kappa}^m,\boldsymbol{\kappa}^+,\boldsymbol{\kappa}^-)}$ (yellow). (C-E) Encoding the additional information necessary for lossless reconstruction of $G$ from $H$, incurs a cost $L(G|H, \mathcal{V}, \mathcal{S}, \Gamma)$ (Eq (8)) that is equal to the sum of three terms for each supernode, corresponding to encoding the labels of the nodes inside the graphlet, i.e., the graphlet's orientation (C), and how the graphlet's nodes are wired to other nodes in $H$ (D,E). (C) Encoding the orientation of a graphlet is equivalent to specifying its automorphism class. For the graphlet shown in the example there are 3 possible distinguishable orientations, leading to a codelength of log 3. (D) Encoding the connections between a simple node and a supernode involves designating to which nodes in the graphlet the in- and out-going edges to the supernode are connected. In this example, there are $\binom{4}{2}$ possible wiring configurations for both the in- and out-going edges, leading to a wiring cost of log 36 (see Eq (9)). (E) Encoding the wiring configuration of the edges from a supernode $i$ to another supernode $j$ involves designating the edges from the group of nodes of supernode $i$ to the group of nodes in $j$ in the bipartite graph composed of the two groups (the edges from $j$ to $i$ are accounted for in the encoding of $j$). Here, there are $\binom{20}{1}$ such configurations, leading to a rewiring cost of log20 bits.

encoding the number $|\Gamma|$. The third term is the cost of encoding the number of times each of the motifs appears, requiring $\log m_{\max}$ bits per motif. The fourth term is the cost of encoding $m_{\max}$.

The second term in Eq (5), $L(H, \phi)$, depends on the base model used to encode $H$. We consider several models and detail their codelengths in the "Base codes and null models" section below.

The third term of Eq (5) is equal to

$$L(\mathcal{V}|H, \mathcal{S}) = \log \binom{N(H)}{|\mathcal{S}|} + \log \frac{|\mathcal{S}|!}{\prod_\alpha m_\alpha!}, \tag{7}$$

where the first part corresponds to the cost of labeling $|\mathcal{S}|$ nodes of $H$ as supernodes—equal to the logarithm of the number of ways to distribute the labels—and the second part corresponds

to the labeling of the supernodes to show which graphlet they each correspond to—equal to the logarithm of the number of distinguishable ways to order $\mathcal{S}$.

The fourth and last term in Eq (5) is given by

$$L(G|H, \mathcal{V}, \mathcal{S}, \Gamma) = \log \frac{N(G)!}{N(H)!} + \sum_{\alpha} m_{\alpha} \log \frac{n_{\alpha}!}{|\mathrm{Aut}(\alpha)|} + \sum_{i_s \in \mathcal{V}} \ell_{\mathrm{rew}}(i_s, H). \tag{8}$$

Here, the first term is the cost of recovering the original node labeling of $G$ from $H$. The second term encodes the orientation of each graphlet to recover the subgraphs found in $G$ (Fig 1C)—for a given graphlet $\alpha$ (consisting of $n_{\alpha}$ nodes) there are $n_{\alpha}!/|\mathrm{Aut}(\alpha)|$ distinguishable orientations, where $|\mathrm{Aut}(\alpha)|$ denotes the size of the automorphism group of $\alpha$. The third term is the *rewiring cost* which accounts for encoding how edges in $H$ involving a supernode are connected to the nodes of the corresponding graphlet. Denoting by $n_s$ the number of nodes in the subgraph $s$ that the supernode $i_s$ replaces, the rewiring cost for one supernode is given by

$$\ell_{\mathrm{rew}}(i_s, H) = \sum_{j \in \mathcal{N}(H) \setminus \mathcal{V}} \log \binom{n_s}{A_{i,j}} \binom{n_s}{A_{ji_s}} + \sum_{j_{s'} \in \mathcal{V}} \log \binom{n_s n_{s'}}{A_{i_s j_{s'}}}, \tag{9}$$

where the first term is the cost for designating which of the possible wiring configurations involving the nodes inside a supernode and adjacent regular nodes corresponds to the configuration found in $G$ (Fig 1D), and the second term is the cost of encoding the wiring configurations for edges from the nodes of the given supernode to the nodes of its adjacent supernodes (Fig 1E).

## Base codes and null models

**The latent graph code.** To encode the latent reduced graph $H$, we use two-part codes of the form $L(H, \phi) = -\log P_{\phi}(H) + L(\phi)$ (Eq (2)), where $L(\phi)$ encodes the parameters of the chosen dyadic random graph model—the model's *parametric codelength*—and $P_{\phi}(H)$ is a uniform probability distribution over a multigraph ensemble conditioned on the value of $\phi$. Note that, while $G$ is a simple graph, the subgraph contractions may generate multiple edges between the same nodes in $H$, which consequently is a multigraph. The models $P_{\phi}$ correspond to maximum entropy microcanonical graph ensembles [56–58], i.e., uniform distributions over graphs with certain structural properties $\phi(H)$, e.g., the node degrees, set to match exactly a given value, $\phi(H) = \phi^*$. The microcanonical distribution is given by

$$P_{\phi}(H) = \begin{cases} \dfrac{1}{\Omega_{\phi}} & \text{for } \phi(H) = \phi^*, \\ 0 & \text{elsewise,} \end{cases} \tag{10}$$

where the normalizing constant $\Omega_{\phi} = |\{H: \phi(H) = \phi^*\}|$ is known as the microcanonical partition function. The codelength for encoding $H$ using the model $P_{\phi}$ can be identified with the microcanonical entropy,

$$-\log P_{\phi}(H) = \log \Omega_{\phi} \equiv S_{\phi}, \tag{11}$$

leading to a total codelength for encoding the model and the reduced graph of

$$L(H, \phi) = S_{\phi(H)} + L(\phi(H)). \tag{12}$$

As base codes we consider four different paradigmatic random graph models, namely the Erdős-Rényi (ER) model, the configuration model (CM), and their reciprocal versions (RER

and RCM, respectively). For the ER model, the parameters are the number of nodes and edges, while the configuration model constrains the nodes' in- and out-degrees and their reciprocal versions additionally constrain the number of reciprocated edges. Both the degree distributions and the edge reciprocity have been found to be significantly non-random in biological networks, and they have been shown to influence the networks' topology and function [8–11, 26, 40, 59–62]. Thus, it is natural to include these features in the base models, and the corresponding models have been widely employed as null models for hypothesis testing-based motif inference [2, 3, 16, 17, 21–23, 25].

Microcanonical models are defined by the features of a graph that they keep fixed [56] (see Eq (10)). We list them for each of the four models below and we give in Table 1 expressions for their entropy $S_\phi$ and their parametric codelength $L(\phi)$ (see Section A in S4 Text for details).

- **The Erdős-Rényi model (ER)** fixes the number of nodes and edges, $\phi = (N, E)$.

- **The configuration model (CM)** fixes the nodes' in- and out-degrees (the number of incoming and outgoing edges), $\phi = (\mathbf{k}^+, \mathbf{k}^-)$.

- **The reciprocal Erdős-Rényi model (RER)** fixes the number of nodes, the number of reciprocal edges, and the number of non-reciprocated edges, $\phi = (N, E_m, E_d)$. Formally, for a

**Table 1. Base- and null-model codelenghts.** The codelength of a model is equal to $L(H, \phi) = S_\phi + L(\phi)$ (Eq (12)), with the entropy $S_\phi$ and the model complexity $L(\phi)$ given by the appropriate expressions in the table. The entropy of multigraph models are given in the first four lines and the entropy of the simple graph models are given in the next four lines. The parametric complexity of the models is the same for multi- and simple graphs and are listed in the following four lines. Finally, expressions for common parametric codelengths are given in the last four lines. For multigraph codes, the asymmetric and symmetric parts of the adjency matrix are denoted by $A_{ij}^{\text{asym}} = \max(A_{ij} - A_{ji}, 0)$ and $A_{ij}^{\text{sym}} = \min(A_{ij}, A_{ji})$, respectively. For reciprocal models (RER and RCM), $E_d = \sum_{i,j} A_{ij}^{\text{asym}}$ is the number of non-reciprocated edges and $E_m = \sum_{i<j} A_{ij}^{\text{sym}}$ is the number of reciprocated edges. For the configuration model (CM), $k_i^+ = \sum_j A_{ij}$ denotes the out-degrees and $k_i^- = \sum_j A_{ji}$ the in-degrees. For the reciprocal CM (RCM), $\kappa_i^+ = \sum_j A_{ij}^{\text{asym}}$ and $\kappa_i^- = \sum_j A_{ji}^{\text{asym}}$ are the non-reciprocated out- and in-degrees, and $\kappa_i^m = \sum_j A_{ij}^{\text{sym}}$ are the reciprocal degrees. (Details can be found in S4 Text).

| Model | Multigraph entropy $S_\phi$ |
|---|---|
| ER | $E \log[N(N-1)] - \log \frac{E!}{\prod_{i \neq j} A_{ij}!}$ |
| RER | $(E_m + E_d) \log[N(N-1)] - \log \frac{(2E_m)!! E_d!}{\prod_{i<j} A_{ij}^{\text{sym}}! A_{ij}^{\text{asym}}! A_{ji}^{\text{asym}}!}$ |
| CM | $\log \frac{E!}{\prod_i k_i^+! k_i^-!} - \sum_{i \neq j} \log A_{ij}!$ |
| RCM | $\log \frac{(2E-1)!!}{\prod_i \kappa_i^m! \kappa_i^+! \kappa_i^-!} - \sum_{i<j} \log A_{ij}^{\text{sym}}! A_{ij}^{\text{asym}}! A_{ji}^{\text{asym}}!$ |

| | Simple graph entropy $S_\phi$ |
|---|---|
| ER | $\log \binom{N(N-1)}{E}$ |
| RER | $\log \frac{[N(N-1)/2]!}{[N(N-1)/2 - E_m - E_d]! E_m! E_d!} + E_d$ |
| CM | $\log \frac{E!}{\prod_i k_i^+! k_i^-!} - \frac{1}{2 \ln 2} \frac{\langle k_i^+{}^2 \rangle \langle k_i^-{}^2 \rangle}{\langle k_i^+ \rangle \langle k_i^- \rangle}$ |
| RCM | $\log \frac{(2E_m)!!}{\prod_i \kappa_i^m!} + \log \frac{E_d!}{\prod_i \kappa_i^+! \kappa_i^-!} - \frac{1}{2 \ln 2} \left( \frac{1}{2} \frac{\langle (\kappa_i^m)^2 \rangle^2}{\langle \kappa_i^m \rangle^2} + \frac{\langle (\kappa_i^+)^2 \rangle \langle (\kappa_i^-)^2 \rangle}{\langle \kappa_i^+ \rangle \langle \kappa_i^- \rangle} + \frac{\langle \kappa_i^+ \kappa_i^- \rangle^2}{\langle \kappa_i^+ \rangle \langle \kappa_i^- \rangle} + \frac{\langle \kappa_i^m \kappa_i^+ \rangle \langle \kappa_i^m \kappa_i^- \rangle}{\langle \kappa_i^m \rangle \langle \kappa_i^+ \rangle} \right)$ |

| | Model complexity $L(\phi)$ |
|---|---|
| ER | $L_{\mathbb{N}}(N) + L_{\mathbb{N}}(E)$ |
| RER | $L_{\mathbb{N}}(N) + L_{\mathbb{N}}(E_m) + L_{\mathbb{N}}(E_d)$ |
| CM | $L_{\text{seq}}(\mathbf{k}^+) + L_{\text{seq}}(\mathbf{k}^-)$ |
| RCM | $L_{\text{seq}}(\boldsymbol{\kappa}^m) + L_{\text{seq}}(\boldsymbol{\kappa}^+) + L_{\text{seq}}(\boldsymbol{\kappa}^-)$ |

*(Continued)*

**Table 1.** (Continued)

| | Parametric codelengths |
|---|---|
| Integer | $L_{\mathbb{N}}(n) = n(n+1)$ |
| Sequence | $L_{\text{seq}}(\mathbf{x}) = \min\{L_U(\mathbf{x}), L_{\lambda=1}(\mathbf{x}), L_{\lambda=1/2}(\mathbf{x})\} + \log 3 + L_{\mathbb{N}}(n)$, with $n = |\mathbf{x}|$ |
| Uniform | $L_U(\mathbf{x}) = n \log (\Delta - \delta + 1) + L_{\mathbb{N}}(\Delta) + L_{\mathbb{N}}(\delta)$, with $n = |\mathbf{x}|$, $\Delta = \max(\mathbf{x})$ and $\delta = \min(\mathbf{x})$ |
| Dirichlet-multinomial | $L_\lambda(\mathbf{x}) = -\log \frac{\Gamma(\Lambda)}{\Gamma(n+\Lambda)} + \frac{\Delta}{\lambda} \log \Gamma(\lambda) - \sum_{\substack{\delta \leq \mu \leq \Delta \\ \mu \in \mathbb{N}}} \log \Gamma[\lambda + \sum_{i=1}^n \delta(x_i, \mu)]$, with $n = |\mathbf{x}|$, $\Delta = \max(\mathbf{x})$, $\delta = \min(\mathbf{x})$, and $\Lambda = (\Delta - \delta + 1)\lambda$ |

simple graph, a (non-)reciprocated edge is conveyed by (a)symmetric entries of the adjacency matrix. That is, an edge $(i, j)$ is reciprocal if $A_{ij} = 1$ and $A_{ji} = 1$ and non-reciprocated if $A_{ij} = 1$ and $A_{ji} = 0$. The definition for multigraphs extends this idea to integer counts by defining the reciprocal part of a multiedge as the minimum of $A_{ij}$ and $A_{ji}$ and the non-reciprocated part as the rest [63] (details can be found in Section A in S4 Text).

- **The reciprocal configuration model (RCM)** fixes the nodes' reciprocal degrees—the number of reciprocal edges each node partakes in—as well as the non-reciprocated in- and out-degrees, $\phi = (\boldsymbol{\kappa}^m, \boldsymbol{\kappa}^+, \boldsymbol{\kappa}^-)$.

The different base models respect a partial order in terms of how random they are, i.e., how large their entropy is (Fig 1B) [56]. We stress that the model with the smallest entropy does not necessarily provide the shortest description of a graph $H$ due to its higher model complexity (see Section A in S4 Text).

**Motif-free reference codes.** To assess the significance of inferred motif sets, we compare the motif-based graph codes to their purely dyadic counterparts. In Table 1, we also list expressions for the entropy of dyadic simple graph codes for the ER, CM, RER, and RCM models (see Section A in S4 Text for details and Section B in S4 Text for a derivation of the entropy of the simple graph RCM). The parametric complexity of the simple graph models are identical to the ones of the multigraph base models. Including these purely dyadic codes in the set of possible models $\mathcal{M}$ ensures that our motif inference is conservative and does not find spurious motifs in random networks (see "Numerical validation" in Results below).

## Optimization algorithm

To infer a motif set, we apply a greedy iterative algorithm that contracts the most compressing subgraph in each iteration. Since the number of $n$-node subgraphs grows super-exponentially in $n$, it is not convenient to consider all subgraphs at once. Thus, we developed a stochastic algorithm that randomly samples a mini-batch of subgraphs in each iteration and contracts the one that compresses the most among these (Fig 2). We give in Algorithms 1–4 pseudocode for its implementation and describe below each of the main steps involved.

**Algorithm 1** Greedy motif inference

```
Input: Graph G, graphlet set Γ, base model P_φ, subgraph minibatch size
B
1: t ← 0
2: H₀ ← G
3: 𝒮₀, 𝒱₀ ← ∅, ∅
4: θ₀ ← (H₀, φ(H₀), 𝒮₀, 𝒱₀, Γ)
5: Θ ← {θ₀}
6: 𝒞 ← SUBGRAPHCENSUS(G, Γ)
7: while 𝒞 is not ∅ do
```

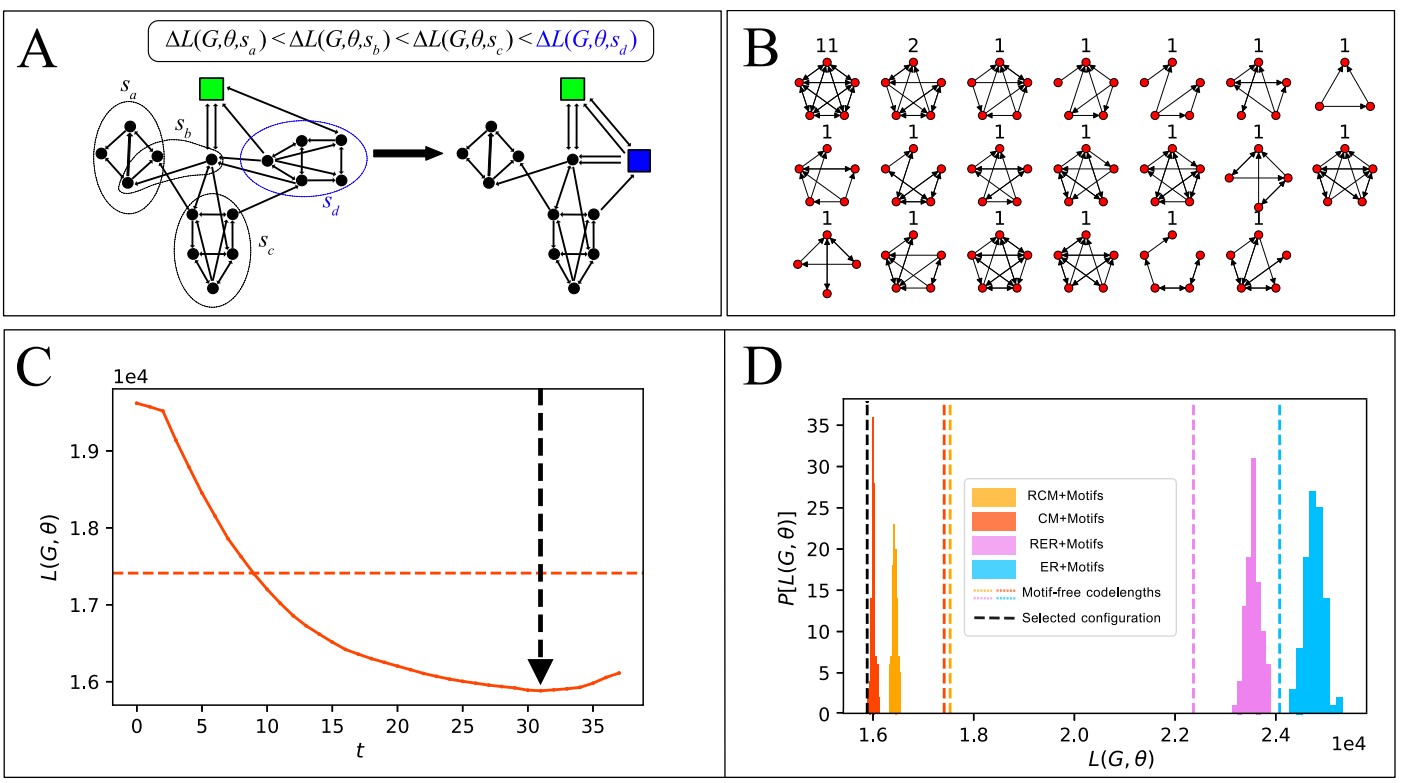

**Fig 2. Greedy optimization algorithm.** (A) Illustration of a single step of the greedy stochastic algorithm. The putative compression $\Delta L(G, \theta, s)$ that would be obtained by contracting each of the subgraphs in the minibatch is calculated, and the subgraph contraction resulting in the highest compression is selected (highlighted in blue). (B) Example of motif set inferred in the connectome of the right hemisphere of the mushroom bodies (MB right) of the *Drosophila* larva. (C) Evolution of the codelength during a single algorithm run. The algorithm is continued until no more subgraphs can be contracted. The representation $\theta^* = \theta_t$ with the shortest codelength is selected; here, after the 31st iteration (indicated by a vertical black dashed line). The horizontal orange dashed line indicates the codelength of the corresponding simple graph model without motifs (see Motif-free reference codes). (D) The algorithm is run a hundred times for each dyadic base model and the most compressing model $\hat{\theta}$ is selected. Histograms represent the codelengths of models with motifs after each run of the greedy algorithm; colors correspond to the different base models (blue: ER model, orange: configuration model, pink: reciprocal ER model, yellow: reciprocal configuration model, see Fig 1B and Table 1); vertical dashed lines represent the codelengths of models without motifs, and the black dashed line indicates the codelength of the shortest-codelength model—here the configuration model with motifs.

```
8:    t ← t + 1
9:    C, B ← SubgraphBatches(B, Γ, C)
10:   α, s_α ← MostCompressingSubgraph(G, B, θ_{t-1})
11:   H_t, S_t, V_t ← SubgraphContraction(H_{t-1}, V_{t-1}, S_{t-1}, α, s_α)
12:   θ_t ← (H_t, φ(H_t), S_t, V_t, Γ)
13:   Θ ← Θ ∪ {θ_t}
14: end while
Output: argmin_{θ∈Θ}{L(G, θ)}
```

**Algorithm 2** Sample subgraph batches

```
1: function SubgraphBatches(B, Γ, C)
2:    B ← ∅
3:    for α ∈ Γ do
4:        B_α ← ∅
5:        while |B_α| < B and |S_α| > 0 do
6:            s_α ← SampleGraphletInstance(C_α)
7:            if NonOverlappingSubgraph(H, s_α) then
8:                B_α ← B_α ∪ {s_α}
9:            else C_α ← C_α \ {s_α}
10:           end if
```

```
11:      end while
12:      𝓑 ← 𝓑 ∪ 𝓑_α
13:      𝓒_α ← 𝓒_α \ 𝓑_α
14:   end for
15:   return 𝓒, 𝓑
16: end function
1: function SAMPLEGRAPHLETINSTANCE(𝓒_α)
2:   return s_α, a subgraph sampled uniformly from 𝓒_α
3: end function
1: function NONOVERLAPPINGSUBGRAPH(H, s_α)
2:   b ← True
3:   for i ∈ s_α do
4:     if i ∉ 𝓝(H) then
5:       b ← False            ▷ Delete node labels of already
           contracted subgraphs
6:     end if
7:   end for
8:   return b
9: end function
```

**Algorithm 3** Find most compressing subgraph.

```
1: function MOSTCOMPRESSINGSUBGRAPH(G, 𝓑, θ)
2:   s* ← argmax_{s∈𝓑}{ΔL(G, θ, s)}       ▷see Section C in S4 Text.
3:   Let α ∈ Γ be the graphlet such that g_α ≅ s*.
4:   return α, s*
5: end function
```

**Algorithm 4** Subgraph contraction

```
1: function SUBGRAPHCONTRACTION(H, 𝒱, 𝒮, α, s_α)
2:   for (i, j) ∈ ℰ(s_α) do
3:     A_{ij}(H) ← 0
4:   end for
5:   𝓝(H) ← 𝓝(H) \ s_α
6:   Let i_α be the label of a new supernode
7:   𝓝(H) ← 𝓝(H) ∪ {i_α}
8:   𝒱 ← 𝒱 ∪ {i_α}
9:   𝒮 ← 𝒮 ∪ {α}
10:   for l ∈ ∂s_α do
11:     A_{i_α l}(H) ← 0
12:     for i ∈ 𝓝(s_α) do
13:       A_{i_α l}(H) ← A_{i_α l}(H) + A_{il}(H)
14:     end for
15:   end for
16:   return H, 𝒱, 𝒮
17: end function
```

**Subgraph census.** (SUBGRAPHCENSUS in Algorithm 1). We first perform a subgraph census to provide a set of lists of the graphlet occurrences in $G$, $\mathcal{C} = \{\mathcal{C}_\alpha : \alpha \in \Gamma\}$ with $\mathcal{C}_\alpha = \{g \equiv G[v] : v \subseteq \mathcal{N} \wedge g \simeq \alpha\}$ (see the "Subgraph census" section above). We consider in the "Results" section below $\Gamma$ to be *all* graphlets of three, four, and five nodes, but any predefined set of graphlets may be specified in the algorithm.

Once the subgraph census is completed, we perform stochastic greedy optimization by iterating the following steps.

**Subgraph sampling.** (SUBGRAPHBATHCHES, Algorithm 2). In each step, the algorithm samples a minibatch of subgraphs, $\mathcal{B}_t$, consisting of $B$ subgraphs per graphlet selected uniformly from $\mathcal{C}$. The SUBGRAPHBATHCHES function also discards subgraphs in $\mathcal{C}$ that overlap with already contracted subgraphs. Indeed, from a biological point of view, overlapping supernodes

correspond to nested circuit motifs, whose significance differs from the standard circuit motifs, where each node is identified with a single unit (e.g., a neuron). Furthermore, this constraint guarantees a faster algorithmic convergence by progressively excluding many subgraphs candidates. The number of subgraphs per graphlet, $B$, is a hyperparameter of the algorithm. We tested different values of $B$ and found similar results for values in the range 10–100 (see S8 Fig).

The check of overlap is performed by a boolean sub-function NonOverlappingSubgraph (see Algorithm 2). It asserts whether a node of a subgraph $s$ is already part of a supernode of $H_{t-1}$.

**Finding the most compressing subgraph.** (MostCompressingSubgraph, Algorithm 3). We calculate for each subgraph $s \in \mathcal{B}_t$ how much it would allow to further compress $G$ compared to the representation of the previous iteration, i.e., the codelength difference $\Delta L(G, \theta_t, s) = L(G, \theta_t) - L(G, \tilde{\theta}_t(s))$, where $\tilde{\theta}_t(s)$ represents the putative parameter set after contraction of $s$ (see Section C in S4 Text for expressions of codelength differences). The subgraph $s^*$ for which $\Delta L$ is maximal is selected for contraction.

**Subgraph contraction.** (SubgraphContraction, Algorithm 4). The reduced graph $H_t$ is obtained by contraction of the subgraph $s^* \equiv s_\alpha$ (isomorphic to the graphlet $\alpha$) in $H_{t-1}$. The subgraph contraction consists of deleting in $H_{t-1}$ the regular nodes and simple edges of $s_\alpha$, and replacing them with a supernode $i_\alpha$ that connects to the union of the neighborhoods of the nodes of $s_\alpha$, denoted $\partial s_\alpha$, through multiedges. We refer to $\partial s_\alpha$ as the subgraph's neighborhood, which, by design, is identical to the supernode's neighborhood. Nodes of $s_\alpha$ that share neighbors will result in the formation of parallel edges, affecting the adjacency matrix according to $A_{i_\alpha j} = \sum_{i \in s_\alpha} A_{ij}$.

**Stopping condition and selection of most compressed representation.** At each iteration $t$, the algorithm generates a compressed version of $G$, parametrized by $\theta_t$. We run the algorithm until no more subgraphs can be contracted, i.e., until there are no more subgraphs that are isomorphic to a graphlet in $\Gamma$ and do not involve a supernode in $H_t$. We then select the representation that achieves the minimum codelength among them (Fig 2C),

$$\theta^* = \text{argmin}\{L(G, \theta_t)\}. \tag{13}$$

**Repeated inferences for each base code.** Since different base models lead to different inferred motif sets (see S2 Fig), we run the optimization algorithm independently for each base model, and since the algorithm is stochastic, we run it 100 times per connectome and base model to gauge its variability and check that the inference is reasonable (Fig 2D). We select the model $\hat{\theta}$ with the shortest codelength among all these, and its corresponding motif set if the best model is one with motifs,

$$\hat{\theta} = \text{argmin}\{L(G, \theta^*)\}. \tag{14}$$

## Datasets

### Artificial datasets.

**Randomized networks.** To quantify the propensity of our approach and of hypothesis testing-based methods to infer spurious motifs (i.e., false positives), we apply them to random networks without motifs. To generate random networks corresponding to the different null models, we apply the same Markov-chain edge swapping procedures [59] used for hypothesis-testing based motif inference (see more details in S1 Text).

**Planted motif model.** To test the ability of our method to detect motifs that genuinely are present in a network (i.e., true positives), we generated random networks according to a

**Table 2. Connectome datasets.** For each connectome, we list its number of non-isolated nodes, $N$, its number of directed edges, $E$, its density $\rho = E/[N(N-1)]$, the features of the most compressing model for the connectome, its compressibility $\Delta L^*$, the difference in codelengths between the best models with and without motifs, $\Delta L_{motifs}$, and the reference to the original publication of the dataset. The absolute compressibility $\Delta L^*$ measures the number of bits that the shortest-codelength model compresses compared to a simple Erdős-Rényi model (Eq (15)). The difference in compression with and without motifs, $\Delta L_{motifs}$, quantifies the significance of the inferred motif sets as the number of bits gained by the motif-based encoding compared to the optimal motif-free, dyadic model. For datasets where no motifs are found, this column is marked as "N/A". All datasets are available at https://gitlab.pasteur.fr/sincobe/brain-motifs/-/tree/master/data.

| Species | Connectome | $N$ | $E$ | $\rho$ | Best model | $\Delta L^*$ | $\Delta L_{motifs}$ | Ref. |
|---|---|---|---|---|---|---|---|---|
| *C. elegans* | Head Ganglia—Hour 0 | 187 | 856 | 0.025 | RCM | 354 | N/A | [39] |
| *C. elegans* | Head Ganglia—Hour 5 | 194 | 1108 | 0.030 | RCM | 494 | N/A | [39] |
| *C. elegans* | Head Ganglia—Hour 8 | 198 | 1104 | 0.028 | RCM | 626 | N/A | [39] |
| *C. elegans* | Head Ganglia—Hour 15 | 204 | 1342 | 0.032 | RCM | 722 | N/A | [39] |
| *C. elegans* | Head Ganglia—Hour 23 | 211 | 1801 | 0.041 | RCM | 957 | N/A | [39] |
| *C. elegans* | Head Ganglia—Hour 27 | 216 | 1737 | 0.037 | RCM | 939 | N/A | [39] |
| *C. elegans* | Head Ganglia—Hour 50 | 222 | 2476 | 0.050 | RCM | 1428 | N/A | [39] |
| *C. elegans* | Head Ganglia—Hour 50 | 219 | 2488 | 0.052 | RCM | 1562 | N/A | [39] |
| *C. elegans* | Hermaphrodite—nervous system | 309 | 2955 | 0.031 | RCM+Motifs | 2167 | **286** | [64] |
| *C. elegans* | Hermaphrodite—whole animal | 454 | 4841 | 0.024 | CM+Motifs | 7605 | **2661** | [65] |
| *C. elegans* | Male—whole animal | 575 | 5246 | 0.016 | CM+Motifs | 8979 | **2759** | [65] |
| *Drosophila* | Larva—left AL | 96 | 2142 | 0.235 | RCM | 1550 | N/A | [66] |
| *Drosophila* | Larva—right AL | 96 | 2218 | 0.244 | RCM | 1527 | N/A | [66] |
| *Drosophila* | Larva—left & right ALs | 174 | 4229 | 0.140 | RCM+Motifs | 4117 | **105** | [66] |
| *Drosophila* | Larva—left MB | 191 | 6449 | 0.167 | CM+Motifs | 8050 | **1369** | [67] |
| *Drosophila* | Larva—right MB | 198 | 6499 | 0.178 | CM+Motifs | 8191 | **1529** | [67] |
| *Drosophila* | Larva—left & right MBs | 387 | 16956 | 0.114 | RCM+Motifs | 23764 | **5348** | [67] |
| *Drosophila* | Larva—motor neurons | 426 | 3795 | 0.021 | CM | 4762 | N/A | [68] |
| *Drosophila* | Larva—whole brain | 2952 | 110140 | 0.013 | RCM+Motifs | 149521 | **28793** | [40] |
| *Drosophila* | Adult—right AL | 761 | 36901 | 0.064 | RCM+Motifs | 76007 | **61** | [69] |
| *Drosophila* | Adult—right LH | 3008 | 100914 | 0.011 | RCM+Motifs | 109473 | **583** | [69] |
| *Drosophila* | Adult—right MB | 4513 | 247863 | 0.012 | RCM+Motifs | 429773 | **13657** | [69] |
| *C. intestinalis* | Larva—whole brain | 222 | 3085 | 0.063 | RCM+Motifs | 3805 | **263** | [70] |
| *P. dumerelii* | Larva—whole brain | 2728 | 11433 | 0.002 | RCM+Motifs | 15733 | **325** | [71] |

*planted motif model* which generates networks with placed motifs by inverting our compression algorithm according to the following steps: (i) generate a random latent multigraph $H$ according to the ER model; (ii) designate at random a predetermined number of the nodes as supernodes; (iii) expand the supernodes by replacing them with the motif of choice, oriented at random and with its nodes wired at random to the supernode's neighbors in $H$.

**Empirical datasets.**

We apply our method to infer microcircuit motifs in synapse-resolution neural connectomes of different small animals obtained from serial electron microscopy (SEM) imaging (see Table 2 for descriptions and references of the datasets). All input raw and processed connectomes can be found in our GitLab project, in the `data` folder(gitlab.pasteur.fr/sincobe/brain-motifs/-/tree/master/data).

## Results

### Numerical validation

To test the validity and performance of our motif inference procedure, we apply it to numerically generated networks with a known absence or presence of higher-order structure in the form of motifs (see "Artificial datasets"in Methods).

**Null networks.** We first test the stringency of our inference method and compare it to classic, hypothesis testing-based approaches. We test whether they infer spurious motifs in random networks generated by the four dyadic random graph models (See "Randomized networks" in the Methods). Since these random networks do not have any non-random higher-order structure, a trustworthy inference procedure should find no, or at least very few, significant motifs.

Hypothesis testing-based approaches to motif inference consist of checking whether each graphlet is significantly over-represented with respect to a predefined null model (we detail the procedure in S1 Text). This approach is highly sensitive to the choice of null model and infers spurious motifs if the chosen null model does not correspond to the generative model (Fig 3A–3D). Nevertheless, when the chosen null model is the generative model, almost no

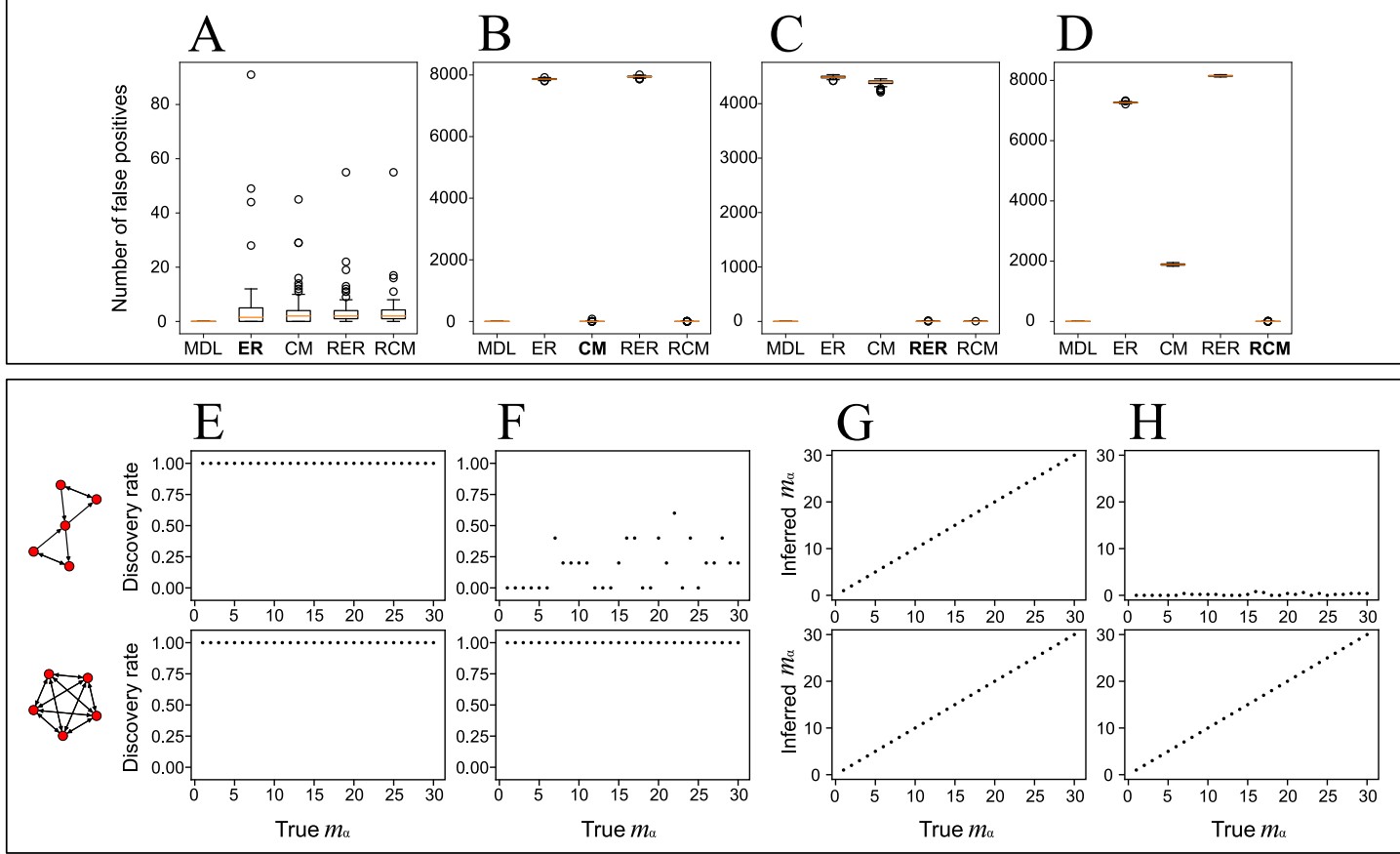

**Fig 3. Performance of compression-based motif inference on numerically generated networks.** (A-D) Number of spurious motifs inferred using our compression-based method with MDL-based model selection and using hypothesis testing with four different null models in random networks generated from the same four null models: (A) the Erdős-Rényi model (ER); (B) the configuration model (CM); (C) the reciprocal ER model (RER); and (D) the reciprocal CM (RCM). The x-axis labels indicate which method was used for motif inference: our method (MDL) or classic hypothesis testing with each of the four null models as reference. The corresponding generative model is highlighted in boldface. To make hypothesis testing as conservative as possible, we applied a Bonferroni correction, which multiplies the raw $p$-values by $|\Gamma| = 9576$ and we set the uncorrected significance threshold to 0.01. The random networks in (A-D) are all generated by fixing the values of each null model's parameters to those of the *Drosophila* larva right MB connectome (e.g., $N = 198$ and $E = 6499$ for the ER model). (E-H) Ability of our method to correctly identify a placed graphlet as a motif as a function of the number of times it is repeated, $m_\alpha$. We show results for two selected 5-node graphlets: an hourglass structure (top row) and a clique (bottom row). The clique is the densest graphlet and is totally symmetric (the number of orientations, i.e., the number of non-automorphic node permutations, is equal to one). The hourglass has intermediary density, $\rho_\alpha = 2/5$, and symmetry, with 60 non-automorphic orientations within a possible range of 1 to 5! = 120. The generated networks in (E-H) contain $N = 300$ nodes and an edge density of either $\rho = E/N(N-1) = 0.025$ (E,G) or $\rho = 0.1$ (F,H). Each point is an average over five independently generated graphs. (E,F) The discovery rate is the estimated probability that the planted motif belongs to the inferred motif set, i.e., $\langle 1 - \delta(m_\alpha, 0) \rangle$. (G,H) Average inferred number of repetitions of the planted motif, $\langle m_\alpha \rangle$.

spurious motifs are found using the approach (Fig 3A–3D). However, since there is no general protocol for the choice of null model in the frequentist approach, this sensitivity to null model choice is a major concern in practice.

By casting motif inference as a model selection problem, our approach allows us to select the most appropriate model, including amongst a selection of null models. In our test, our approach consistently selects the true generative model for the networks, i.e., one of the four null models, and thus does not infer any spurious motifs (Fig 3A–3D).

**Planted motifs.**   To evaluate the efficiency of our method in finding motifs that are present in a network, we apply it to synthetic networks with planted motifs (see "Planted motif model" in the Methods).

We show in Fig 3E–3H the ability of our algorithm to identify a motif (Fig 3E and 3G) and its occurrences (Fig 3F and 3H) in numerically generated networks as a function of the number of times the motif is repeated in the network. We show in S3–S6 Figs a more in-depth analysis including additional motifs, different network sizes, and an extended range of network densities. The performance of the algorithm is affected by both the frequency of the planted motif (Fig 3E–3H) and its topology, with denser motifs generally being easier to identify (Fig 3E–3H, see also S3 and S6 Figs). The size of the network does not have a significant effect on our ability to detect motifs, but its edge density does (compare S3 and S4 Figs to S5 and S6 Figs). The latter is expected since motifs whose density differs significantly from the network's average density are easier to identify than motifs with a similar density. This is similar to hypothesis testing-based approaches based on graphlet frequencies where dense motifs tend to be highly unlikely under the null model and thus easier to detect. However, we stress that our method does not rely on the same definition of significance—compression instead of over-representation—so the motifs that are easiest to infer are not necessarily the same with the different approaches (S2 Fig).

## Neural connectomes

We apply our method to infer circuit motifs in structural connectomes and characterize the regularity of the connectivity of synapse-resolution brain networks of different species at different developmental stages (see Table 2). We consider boolean connectivity matrices that represent neural wiring as a binary, directed network where each node represents a neuron and an edge represents the presence of synaptic connections from one pre-synaptic neuron to a post-synaptic neuron. To keep in line with the usual definition of a motif, we exclude self-connections of neurons onto themselves, but they can be included if one wants to investigate such motifs.

We measure the compressibility of a connectome $G$ as the difference in codelength between its encoding using a simple Erdős-Rényi model, i.e., encoding the edges individually, and its encoding using the most compressing model,

$$\Delta L^* = L(G, (N, E)) - L(G, \theta^*). \tag{15}$$

As Fig 4 and Table 2 show, all the empirical connectomes are compressible, confirming their non-random structure (see S7 Fig for a comparison of all the models considered). Significant higher-order structures in the form of motifs are found in all the whole-CNS and whole-nervous-system connectomes studied here (Fig 4A) as well as in many connectomes of individual brain regions (Fig 4B and 4C). Besides motifs, we find significant non-random degree distributions of the nodes in all connectomes (Fig 4). This is consistent with node degrees being a salient feature of many biological networks, including neuronal networks [2]. Reciprocal connections are also a significant feature of almost all connectomes studied, in alignment

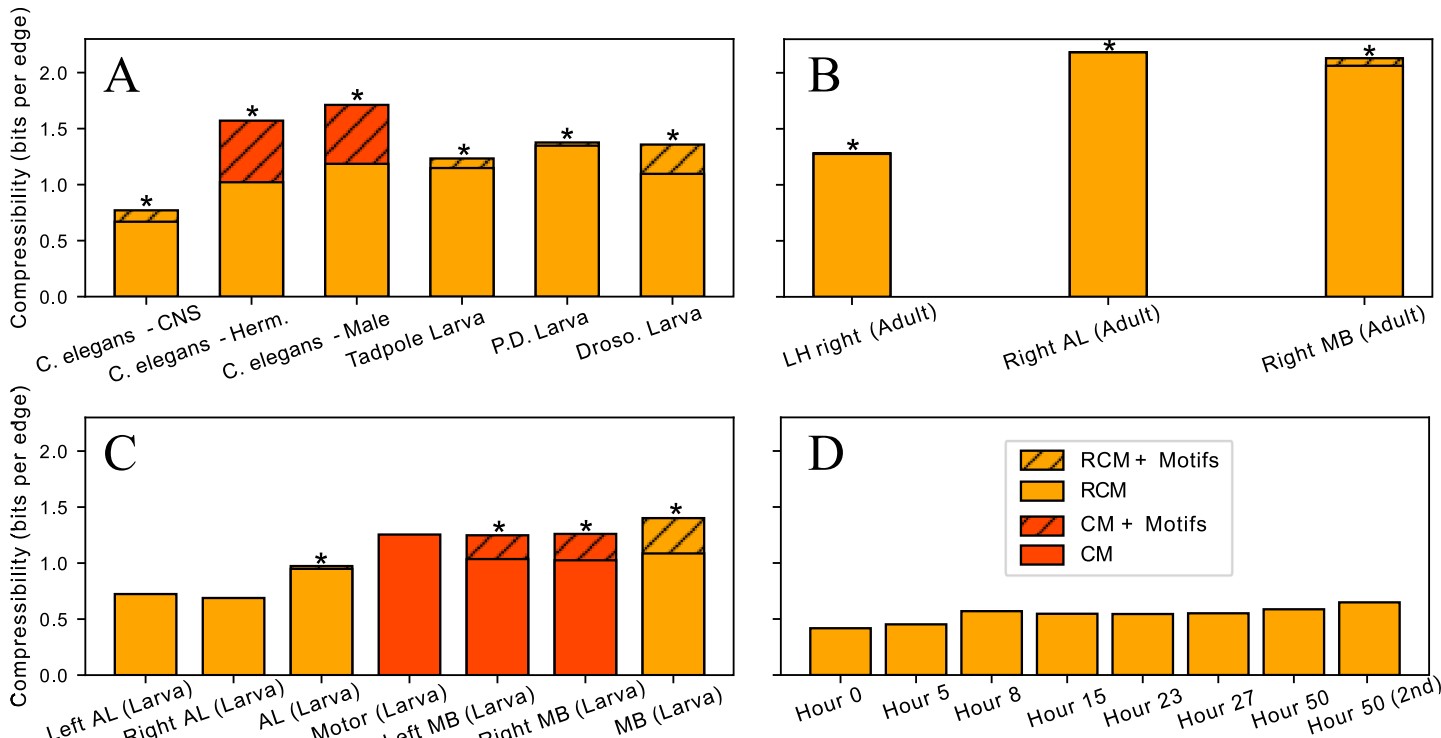

**Fig 4. Compressibility of neural connectomes.** Compressibility (measured in number of bits per edge in the network) $\Delta L^*/E$ of different connectomes as compared to encoding the edges independently using the Erdős-Rényi simple graph model (see Table 1). Two types of models are shown for the datasets: the best simple network encoding and the best motif-based encoding when this compresses more than the simple encoding. Asterisks highlight connectomes where motifs permit a higher compression than the reference models. (A) Whole-CNS and whole-animal connectomes. (B) Connectomes of three different regions of the adult *Drosophila* right hemibrain. Note that while the relative increase in compressibility of these connectomes obtained using motifs is relatively small, the motifs are highly significant due to the large size of these connectomes (Table 2). (C) Connectomes of different brain regions of first instar *Drosophila* larva. (D) Connectomes of *C. elegans* head ganglia at different developmental stages, from 0 hours to 50 (adult). While no higher-order motifs are found, the compressibility increases with maturation (and thus the size) of the connectome.

with empirical observations from *in vivo* experiments [40, 60, 61, 72, 73] where modulation of neural activity is often implemented through recurrent patterns. Note that reciprocal connections are often considered a two-node motif. We chose to encode it as a dyadic feature of the base model since this is more efficient and allows for a higher compression, but it is entirely possible to encode them as graphlets by allowing also two-node graphlets as supernodes in the reduced graph (instead of restricting to 3–5 node graphlets as we did here).

For several smaller, regional connectomes, we do not find statistical evidence for higher-order motifs (Fig 4C and 4D), indicating the absence of significant higher-order circuit patterns (i.e., involving more than two neurons) in these connectomes. Note that network size did not have a significant effect on motif detectability in our numerical experiments above (see S3–S6 Figs), so the absence of motifs in these connectomes are likely due to their structural particularities rather than simply their smaller size. In particular, we do not find evidence for motifs in the *C. elegans* head ganglia (brain) connectomes at any developmental stage (Fig 4D). Note, however, that we do detect significant edge and node features (as encoded by the reciprocal configuration model), highlighting the non-random distribution of neuron connectivity and the importance of feedback connections in these connectomes. Furthermore, we do find higher-order motifs in the more complete *C. elegans* connectomes that also include

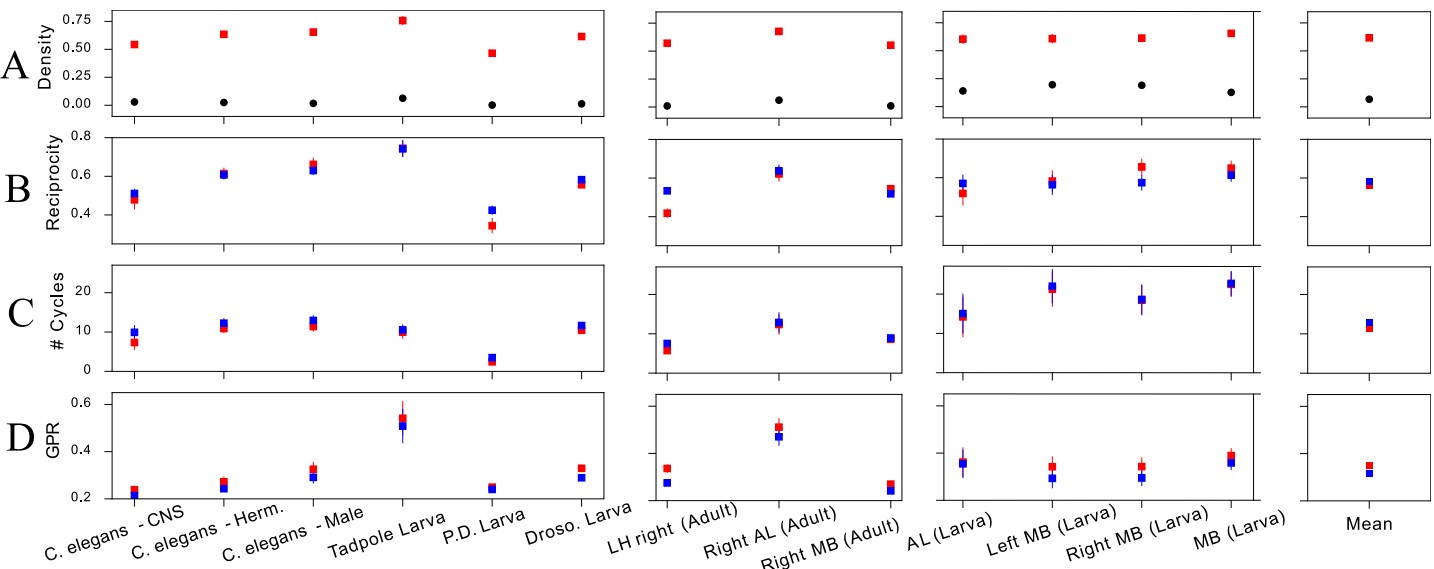

**Fig 5. Topological properties of motif sets.** Graph measures averaged over the inferred graphlet multiset, $\mathcal{S}$, i.e., for a network measure $\varphi$, one point corresponds to the quantity $\mu_\varphi(\mathcal{S}) = \sum_{\alpha \in \mathcal{S}} \varphi(\alpha)/|\mathcal{S}|$. The density (A), reciprocity (B) and number of cycles (C) and are standard properties of directed networks [75]. The graph polynomial root (D) measures the structural symmetry of the motifs [74]. Details can be found in S6 Text. Red squares indicate averages over the connectomes' inferred motif sets. Blue squares are reference values, computed from average over randomized graphlets with their density conserved. To obtain the fixed-density references per motif set, we generate for each graphlet a collection of a hundred randomized configurations sharing the same density. The black dots of panel (A) show the connectomes' global density.

sensory and motor neurons (Fig 4A), in line with what was found earlier using hypothesis-testing based motif mining [16, 65].

To study the structural properties of the inferred motif sets, we computed different average network measures of the motifs of each connectome (see definitions in S6 Text). The density of inferred motifs is much higher than the average density of the connectome (Fig 5A). While the density of motifs is high for all connectomes, it does vary significantly between them in a manner that is seemingly uncorrelated with the average connectome density. The motifs' high density means that half of their node pairs or more are connected on average, which would lead to high numbers of reciprocal connections even if the motifs were wired at random. We indeed observe a high reciprocity of connections in the inferred motifs, and that this reciprocity is in large part explained by their high average density (Fig 5B), although we do observe significant variability and differences from this random baseline. The average number of cycles in the motifs is, on the other hand, in general completely explained by the motifs' high density (Fig 5C). To probe the higher-order structure of the inferred motifs we measure their symmetry as measured by the graph polynomial root (GPR) [74]. As Fig 5D shows, the motifs are on average more symmetric than random graphlets of the same density even if the individual differences are often not significant. Thus, of the four aggregate topological features we investigated, the elevated density is the most salient feature of the motif sets. This does not exclude the existence of salient (higher-order) structural particularities of the motifs beyond their high density, only that such features are not captured well by these simple aggregate measures.

Even though the inferred motif sets are highly diverse, we observe that several motifs are found in a large fraction of the connectomes (Fig 6A). The same motifs also tend to be among the most frequent motifs, i.e., the ones making up the largest fraction of the inferred motif sets on average (Fig 6B). These tend to be highly dense graphlets, with the two most frequent motifs

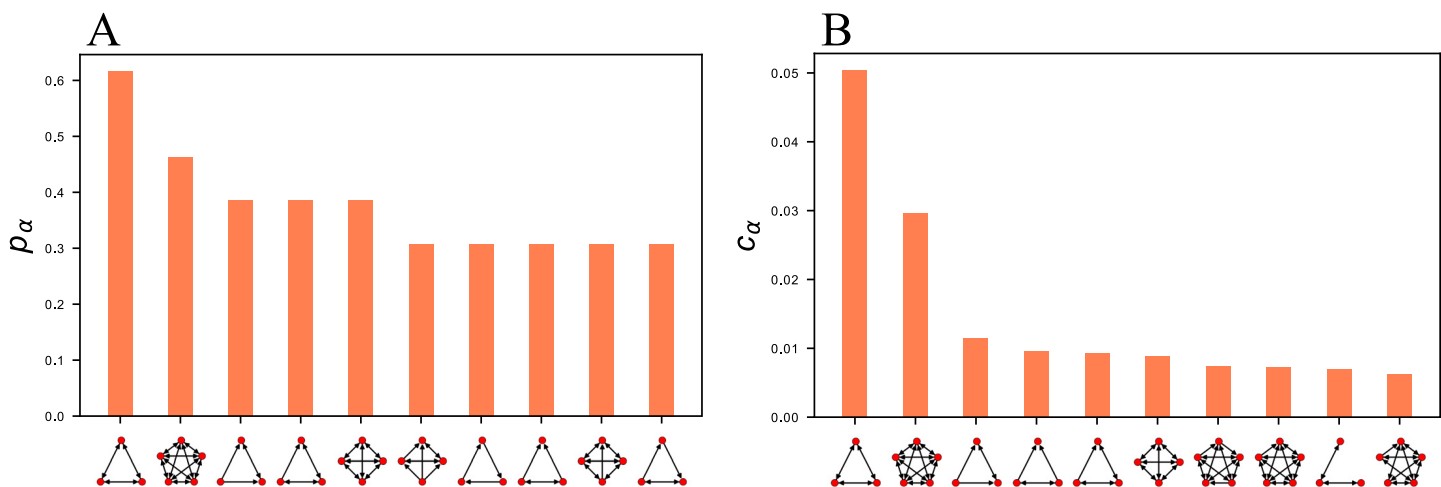

**Fig 6. Connectomes share common motifs.** Most frequently appearing motifs in the motif sets inferred for all connectomes. (A) Most frequently found motifs: fraction of connectomes in which each motif is found, $p_\alpha = \frac{1}{|\mathcal{G}|} \sum_{G \in \mathcal{G}} (1 - \delta_{m_\alpha(G),0})$. (B) Most repeated motifs: average graphlet concentration $c_\alpha = \frac{1}{|\mathcal{G}|} \sum_{G \in \mathcal{G}} \frac{m_\alpha(G)}{\sum_\alpha m_\alpha(G)}$.

being the three and five node cliques, which are each found in roughly half of the connectomes and are also the most frequent motifs in the motif sets on average. The ten most frequently found motifs (Fig 6A) and the most repeated motifs (Fig 6B) do not perfectly overlap, though six of the ten motifs are the same between the two lists.

## Discussion

We have developed a methodology to infer sets of network motifs and evaluate their collective significance based on lossless compression. Our approach defines an implicit generative model and lets us cast motif inference as a model selection problem through the MDL principle. It overcomes several common limitations of traditional hypothesis testing-based methods, which are unable to compare the significance of different motifs and have difficulties dealing with multiple testing, correlations between motif counts, the evaluation of low $p$-values, and the often ill-defined problem of choosing the proper null model to compare against.

Our compression-based methodology accounts for multiple testing and correlations between motifs, and it does not rely on approximations of the null distribution of a test statistic. Note that such approximations are generally necessary for statistical hypothesis testing to be computationally feasible. For example, there are about 10 000 possible five-node motifs, so to control for false positives using the Bonferroni correction, raw $p$-values must be multiplied by 10 000. Thus, one needs to be able to reliably estimate raw $p$ values smaller than $5 \cdot 10^{-6}$ to evaluate significance at a nominal level of 0.05. To obtain an exact test, we must generate of the order of a million random networks and perform a subgraph census of each, a typically unfeasible computational task. Furthermore, constrained null models are hard to sample uniformly [30], and even in models that are simple enough for the Markov chain edge swap procedure to be ergodic, correlations may persist for a long time, inducing an additional risk of spurious results [28, 29].

Our method furthermore allows us to infer not only significant motif sets but also compare and rank the significance of different motifs and sets of motifs and other network features such as node degrees and reciprocity of edges. It thus overcomes the need for choosing the null model a priori, which leads to spurious motifs if this choice is not appropriate.

Note that while our method enables statistically grounded inference of motif sets, it does not provide an estimate of their intrinsic statistical variability since it relies on a greedy optimization algorithm—in the language of Bayesian inference, inferred motif sets correspond to maximum a posteriori estimates. This variability could in principle be estimated via Markov chain Monte Carlo (MCMC) sampling around the optimum motif set, but the development of an efficient MCMC algorithm is an open problem. Thus, for the time being, the variability can only be assessed experimentally by comparing multiple measurements.

Our method is conceptually close to the subgraph covers proposed in [76] which models a graph with motifs as the projection of overlapping subgraphs onto a simple graph and relies on information theoretic principles to select an optimum cover. That approach modeled the space of subgraph covers as a microcanonical ensemble instead of the observed graph directly. This makes it harder to fix node- and edge-level features such as degrees and reciprocity since these are functions of the cover's latent variables [77], although progress in inferring such features has recently been made [78]. We instead based our methodology on subgraph contractions as proposed in [34], whose approach we extended to allow for collective inference of motif sets and selection of base model features. In particular, we let the number of distinct graphlets be free in our method, instead of being limited to one; to deal with the problem of selecting between thousands of graphlets, we developed a stochastic greedy algorithm that selects the most compressing subgraph at each step; we simplified the model for the reduced graph by using multigraph codes, avoiding multiple prequential plug-in codes to account for parallel edges and providing exact codelengths; and we developed two new base models to account for reciprocal edges.

We emphasize that the method we extended [34] and ours are not the first ones to rely on the MDL principle for network pattern mining (see, e.g., the survey in [79]). The SUBDUE [80] and VoG [81] algorithms in particular are precursors of our work, though their focus was on graph summarization rather than motif mining. The SUBDUE algorithm [80] deterministically (but not optimally) extracts the graphlet that can compress a fixed encoding of the adjacency matrix and edge list when a sample of isomorphic (and quasi-isomorphic) subgraphs are contracted. The VoG algorithm [81] uses a set of graphlet types, e.g., cliques or stars, and looks for the set of subgraphs (belonging exactly or approximately to these graphlet types) that best compresses a fixed encoding of the adjacency matrix; the latter being distinct from the one used in SUBDUE. These algorithms differ conceptually from ours in focusing not on motif mining but on more specific regularities for the problem of graph summarization. Their advantage is mainly computational as their implementations scale better with the input graph size. While being computationally more expensive, our approach does not impose or reduce a graphlet dictionary and the representation of the reduced graph is not constrained by a specific functional form.

Exponential random graph models (ERGMs) provide another generative framework for the problem of inferring important subgraphs of a network [82, 83]. Different from our approach, ERGMs generally rely on global graphlet counts and not on contracting specific subgraphs. This tends to make them unstable for general graphlets, making them hard to fit, due to issues of near-degeneracy [83, 84]. This severely conditions the flexibility of ERGMs for motif inference since only a constrained set of particular combinations of motifs are known to ensure convergence of model fits [32, 85, 86].

We applied our approach to uncover and characterize motifs and other structural regularities in synapse-resolution neural connectomes of several species of small animals. We find that the connectomes contain significant structural regularities in terms of a high number of feedback connections (high reciprocity), non-random degrees, and higher-order circuit motifs. In some smaller connectomes we do not find significant evidence for higher order motifs. This is

in particular the case for connectomes of the head ganglia of *C. elegans*, both at maturity and during its development. We still find significant reciprocity and non-random degrees in these connectomes though, confirming the fundamental importance of these measures in biological connectomes. A high reciprocity in particular translates to a large number of feedback connections in the animals' neural networks, a feature whose biological importance has frequently been reported [26, 40, 60–62].

The functional importance of higher-order motifs is less well known, but dense subgraphs are known to have an impact on information propagation in a network [87] and several circuit motifs have been proposed to carry out fundamental computations (e.g., feedforward and feedback regulation [3, 16, 25], cortical computations [88–90], predictive coding [91], and decision making [26]). With the advent of synaptic resolution connectomes, the stage is now set for testing these hypotheses and comparing the structural characteristics of different networks with robust statistical tools such the method we introduced here. While we demonstrated our methodology's ability to detect the most significant circuit patterns in a network among all possible graphlets, it may directly be applied to test for the presence of pre-specified motifs such as the ones cited above by simply changing the graphlet set to include only those circuits.

The mere presence of statistically regular features does not reveal their potential function, nor their origin [92]. These questions must be explored through computational modeling and, ultimately, biological experiments [24–26, 93]. In this aspect our methodology offers an additional advantage over frequency-based methods since it infers not only motifs but also their localization in the network, making it possible to better inform physical models of circuit dynamics and to test their function directly in *in vivo* experiments.

The compressibility of all the neural connectomes investigated here can be seen as a manifestation of the *the genomic bottleneck principle* [94], which states that the information stored in an animal's genome about the wiring of its neural connectome must be compressed or the quantity of information needed to store it would exceed the genome's capacity. Note however that the codelengths needed to describe the connectomes we infer are necessarily upper bounds on the actual codelengths needed to encode the neural wiring blueprints. First, our model is a crude approximation to reality, and a more realistic (and thus more compressing) model would incorporate the physical constraints on neural wiring such as its embedding in 3D space, steric constraints, and the fact that the nervous system is the product of morphogenesis. Second, our code is lossless, which means we perfectly encode the placement of each link in the connectome, while the wiring of neural connections may partially be the product of randomness. Thus a lossy encoding would be a more appropriate measure of a connectome's compressibility [95] but it introduces the difficulty of defining the appropriate distortion measure. Third, subgraph census quickly becomes computationally unfeasible for larger motifs, which generally limits the size of motifs we can consider to less than ten nodes. Allowing for overlapping contractions could be a way to infer larger motifs as combinations of smaller ones (similar to [96]).

We proposed four different base models for our methodology, which allows to select and constrain the important edge- and node-level features of reciprocity and degrees in our model. It is straightforward to incorporate additional base models as long as their microcanonical entropy can be evaluated efficiently. We envisage two important extensions to the base models. First, block structure, which may be incorporated as a stochastic block model [97, 98], is ubiquitous in biological and other empirical networks and has been shown to have an important impact on signal propagation [99]. Second, the network's embedding in physical space, as modelled using geometric graphs or other latent space models [100, 101], is also meaningful. It

should matter for neuronal networks due to considerations such as wiring cost [90], signal latency [90], and steric constraints [90].

Our approach contributes both to the burgeoning field of higher-order networks [15] and to the growing push towards principled statistical inference of network data [102] by providing a robust generative framework for motif inference. The field of statistical network analysis is still in its infancy and much work is still needed to make inference methods more robust. Here, we have for example not considered the problems of noisy data and incomplete sampling [103] which can influence the apparent structure and dynamic of network data in complex ways [103–106]. It should be interesting to extend statistical inference to non-local higher order structures, such as symmetry-group based structures [107] or, e.g., hierarchically nested motifs which might be incorporated in a similar manner to the recent hierarchical extensions of stochastic block models [97, 108]. A common barrier to the development of principled statistical inference of many network models is that they do not admit easily tractable likelihoods. This is in particular the case for many higher-order models, such as the one of [107], and, more famously, for the small world model of Watts and Strogatz [4] and the preferential attachment model of Barabási and Albert [8]. Simulation-based inference [109] provides a promising framework for bridging the gap between such models and statistical inference [110].

## Supporting information

**S1 Text. Classic motif mining based on hypothesis testing.**
(PDF)

**S2 Text. Computational time & memory of the motif-based inference. Table A in S2 Text** recapitulates the computational costs for the motif-based inference when the maximal subgraph size is 4. **Table B in S2 Text** recapitulates the same costs when the maximal subgraph size is 5.
(PDF)

**S3 Text. Subgraph census: Dealing with lists of induced subgraphs.**
(PDF)

**S4 Text. Graph codelengths and subgraph contraction costs. Table A in S4 Text** lists the parameters of the dyadic graph models and some important relationships between them.
(PDF)

**S5 Text. Generating random graphs from the null models.**
(PDF)

**S6 Text. Measures of graphlet topology. Fig A in S6. Text. Distribution of graph polynomial root (GPR) values of all 3–5-node graphlets.** The minimum value of the GPR, for five-node graphlets, is 1/5. It would be 0 in an infinite, maximally asymmetric graph, e.g., one where the automorphism group is a singleton. A GPR of 1, i.e., its maximum value for any graph size, represent maximally symmetric graphs, e.g., cliques. The symmetry of inferred motif sets in Fig 5 in the "Results" section should be interpreted knowing that the GPR is bounded between 0.2 and 1.
(PDF)

**S1 Fig. Differences in the motifs inferred using hypothesis testing when using different null models.** (A) Number of apparent motifs inferred in the *Drosophila* larva right mushroom body connectome when using each of the four null models. Note in particular that even though

the reciprocal models are strictly more constrained than their directed counterparts, more motifs are found with these null models than with the less constrained ones. (B) Overlap (Jaccard index) between the inferred graphlets using the different null models. (C) Per null model, fraction of uniquely found motifs compared to another null model. Formally, denoting by $\mathcal{M}_i$ the motif set WRT the null model in the $i$-th row, and by $\mathcal{M}_j$ the motif set WRT the null model in the $j$-th column, matrix entries are computed as $|\mathcal{M}_i \setminus \mathcal{M}_j|/|\mathcal{M}_i|$. A low ratio indicates that $\mathcal{M}_j$ contains most of $\mathcal{M}_i$, while a high ratio expresses strong dissimilarities between the two emerged motif sets.
(EPS)

**S2 Fig. Different motif sets obtained with the four base models.** Inferred motif sets of the best model for the right hemisphere of the *Drosophila* larva MB connectome. In this specific application, over all inferences across base models, the configuration model has the lowest codelength. We observe a particularly clear distinction in the main types of motifs between Erdős-Rényi-like and configuration-like models.
(EPS)

**S3 Fig. Probability of correctly identifying the embedded motif in the planted motif model ($N$ = 300).** Probability of the inferred motif set containing at least one repetition of the true planted motif as a function of the number of times the motif is planted for five different planted motifs and for different network densities. The generated networks contain $N$ = 300 nodes and the edge density ranges from $\rho$ = 0.01 (leftmost) to $\rho$ = 0.1 (rightmost). Each point is an average over five independently generated graphs. Note that the maximum number of motifs that can be inserted depends both on the number of nodes in the network and on the networks density, as well as that of the motif; hence the range of the x-axis can vary.
(EPS)

**S4 Fig. Number of occurrences of the planted motif inferred ($N$ = 300).** The number of insertions in the generated graphs is plotted on the x-axis, and the inferred number, averaged over five independent graphs, on the y-axis. The generated networks contain $N$ = 300 nodes and the edge density ranges from $\rho$ = 0.01 (leftmost) to $\rho$ = 0.1 (rightmost). Each point is an average over five independently generated graphs. Note that the maximum number of motifs that can be inserted depends both on the number of nodes in the network and on the networks density as well as that of the motif; hence the range of the x-axis can vary.
(EPS)

**S5 Fig. Probability of correctly identifying the embedded motif in the planted motif model ($N$ = 100).** Probability of the inferred motif set containing at least one repetition of the true planted motif as a function of the number of times the motif is planted for five different planted motifs and for different network densities. The generated networks contain $N$ = 100 nodes and the edge density ranges from $\rho$ = 0.01 (leftmost) to $\rho$ = 0.1 (rightmost). Each point is an average over five independently generated graphs. Note that the maximum number of motifs that can be inserted depends both on the number of nodes in the network and on the networks density as well as that of the motif; hence the range of the x-axis can vary.
(EPS)

**S6 Fig. Number of occurrences of the planted motif inferred ($N$ = 100).** The number of insertions in the generated graphs is plotted on the x-axis, and the inferred number, averaged over five independent graphs, on the y-axis. The generated networks contain $N$ = 100 nodes and the edge density ranges from $\rho$ = 0.01 (leftmost) to $\rho$ = 0.1 (rightmost). Each point is an average over five independently generated graphs. Note that the maximum number of motifs

that can be inserted depends both on the number of nodes in the network and on the networks density as well as that of the motif; hence the range of the x-axis can vary.
(EPS)

**S7 Fig. Compressibility per edge of the connectomes obtained with the different base models, with and without motifs.** Difference in codelength between the simple Erdős-Rényi (ER) model and each of the other seven models (RER: reciprocal ER model, CM: configuration model, RCM: reciprocal configuration model, ER+Motifs: ER base model with motifs, RER+Motifs: reciprocal ER base with motifs, CM+Motifs: configuration model with motifs, RCM+Motifs: reciprocal configuration model with motifs).
(EPS)

**S8 Fig. Dependence of the optimum model on the batch size.** Mean codelength of the inferred model (± SD) for different minibatch sizes *B*, where *B* is the number of occurrences of each graphlet sampled. The inference is performed on the *Drosophila* larva right MB and run 100 times independently for each *B* value.
(EPS)

## Acknowledgments

We acknowledge the help of the HPC Core Facility of the Institut Pasteur for this work.

## Author Contributions

**Conceptualization:** Alexis Bénichou, Christian L. Vestergaard.

**Data curation:** Alexis Bénichou, Jean-Baptiste Masson, Christian L. Vestergaard.

**Formal analysis:** Alexis Bénichou, Christian L. Vestergaard.

**Funding acquisition:** Jean-Baptiste Masson, Christian L. Vestergaard.

**Investigation:** Alexis Bénichou.

**Methodology:** Alexis Bénichou, Christian L. Vestergaard.

**Project administration:** Jean-Baptiste Masson, Christian L. Vestergaard.

**Resources:** Jean-Baptiste Masson, Christian L. Vestergaard.

**Software:** Alexis Bénichou.

**Supervision:** Jean-Baptiste Masson, Christian L. Vestergaard.

**Validation:** Alexis Bénichou, Christian L. Vestergaard.

**Visualization:** Alexis Bénichou.

**Writing – original draft:** Alexis Bénichou, Christian L. Vestergaard.

**Writing – review & editing:** Alexis Bénichou, Jean-Baptiste Masson, Christian L. Vestergaard.

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
