## [Decision Letter · Decision Letter 0]

8 Apr 2024

Dear Dr. Vestergaard,

Thank you very much for submitting your manuscript "Compression-based inference of network motif sets" for consideration at PLOS Computational Biology.

As with all papers reviewed by the journal, your manuscript was reviewed by members of the editorial board and by several independent reviewers. In light of the reviews (below this email), we would like to invite the resubmission of a revised version that takes into account the reviewers' comments.

We cannot make any decision about publication until we have seen the revised manuscript and your response to the reviewers' comments. Your revised manuscript is also likely to be sent to reviewers for further evaluation.

Sincerely,

Fabrizio De Vico Fallani

Guest Editor

PLOS Computational Biology

Thomas Serre

Section Editor

PLOS Computational Biology

Reviewer's Responses to Questions

**Comments to the Authors:**

Reviewer #1: see attached file

Reviewer #2: In the manuscript "Compression-based inference of network motif sets", A. Benichou and colleagues propose a framework for motif mining based on lossless network compression using subgraph contractions. They applied this framework to synthetic benchmark and neural connectomes, which allow the evaluation of the collective significance of motif sets in terms of a null model inferred from the data.

Altogether, this represents a valuable contribution to the network and neuroscientific communities. However, certain sections need rephrasing and clarification before acceptance. Below are my comments and suggestions:

- While most of the manuscript is well-written, the abstract, introduction and part of the method could benefit from being more concise. Also, in the introduction and abstract, explicitly mentioning that the approach was first assessed on numerical benchmarks and then applied to neural connectomes would enhance clarity.

- I believe that it would be important to expand on the time and space complexity of the proposed approach, including information about some of the cases reported in the manuscript (e.g. the authors mention the need of storing 1.3 TB of data for the Drosophila connectome). This will offer valuable insights into usability and complexity in practical cases.

- Although the code and scripts used in the manuscript are available on GitHub, providing additional documentation and example cases on how to use the code and perform simple analyses would increase its usability among scientists working on these topics.

- Even though the approach does not allow for overlapping contractions, the motifs identified in the C.elegans connectome could be also linked to recent works on symmetric motifs in C.elegans (e.g. https://www.nature.com/articles/s41467-019-12675-8). Are there any commonalities in the motifs obtained in the manuscript and the symmetric building blocks reported in the paper? If so, this could be useful to strengthen the importance of some of motifs identified in the empirical datasets.

- In addition to the recent work on bayesian stochastic block modeling, the authors might consider exploring parallels with the hierarchical block model discussed in another study (https://journals.aps.org/prx/abstract/10.1103/PhysRevX.4).

Minor Comments:

- Caption Figure 1: "In this example," instead of "In this eample,"

- Figure 2D: Adding a legend explaining the meaning of the colors would improve interpretation.

- Line 210: Correcting "m_{max.}" to "m_{max}"

- Lines 211-213: The paragraph appears to be redundant or repeated.

- Figure labels in Figure 3: Placing them on the side of the figure, as done in other figures, would be ideal.

**Have the authors made all data and (if applicable) computational code underlying the findings in their manuscript fully available?**

Reviewer #1: Yes

Reviewer #2: Yes

PLOS authors have the option to publish the peer review history of their article (what does this mean?). If published, this will include your full peer review and any attached files.

Reviewer #1: No

Reviewer #2: No
---

## [Decision Letter · Decision Letter 1]

4 Sep 2024

Dear Dr. Vestergaard,

We are pleased to inform you that your manuscript 'Compression-based inference of network motif sets' has been provisionally accepted for publication in PLOS Computational Biology.

Best regards,

Fabrizio De Vico Fallani

Guest Editor

PLOS Computational Biology

Thomas Serre

Section Editor

PLOS Computational Biology

Reviewer's Responses to Questions

**Comments to the Authors:**

Reviewer #1: Than you for addressing all the comments and concerns of the reviewers in a constructive manner. I think that this version of the manuscript is an improvement with respect to the previous one that is suitable for publication in PLoS Computational Biology.

Reviewer #2: The authors have successfully addressed all of my concerns. The revised version of the manuscript shows significant improvement compared to the previous version. The manuscript is now much more concise and contains the necessary information to fully comprehend the work they have conducted. As such, I have no further comments on this version of the manuscript.

**Have the authors made all data and (if applicable) computational code underlying the findings in their manuscript fully available?**

Reviewer #1: Yes

Reviewer #2: None

PLOS authors have the option to publish the peer review history of their article (what does this mean?). If published, this will include your full peer review and any attached files.

Reviewer #1: No

Reviewer #2: No

---

## [Editor Report · Acceptance letter]

25 Sep 2024

PCOMPBIOL-D-23-01928R1 

Compression-based inference of network motif sets

Dear Dr Vestergaard,

I am pleased to inform you that your manuscript has been formally accepted for publication in PLOS Computational Biology. Your manuscript is now with our production department and you will be notified of the publication date in due course.

With kind regards,

Anita Estes
